# Motivation and Lifestyle-Related Changes among Participants in a Healthy Life Centre: A 12-Month Observational Study

**DOI:** 10.3390/ijerph19095167

**Published:** 2022-04-24

**Authors:** Cille H. Sevild, Christopher P. Niemiec, Sindre M. Dyrstad, Lars Edvin Bru

**Affiliations:** 1Department of Public Health, University of Stavanger, 4021 Stavanger, Norway; christopher.niemiec@rochester.edu (C.P.N.); sindre.dyrstad@uis.no (S.M.D.); 2Research Unit, Centre for Health Promotion, 4068 Stavanger, Norway; 3Department of Psychology, University of Rochester, Rochester, NY 14642, USA; 4Centre for Learning Environment, University of Stavanger, 4021 Stavanger, Norway; edvin.bru@uis.no; 5Department of Education and Sport Science, University of Stavanger, 4021 Stavanger, Norway

**Keywords:** autonomous motivation, body composition, lifestyle change, perceived competence, physical health, mental health, self-determination theory

## Abstract

Healthy Life Centers (HLCs) have been established throughout Norway to support lifestyle changes and promote physical and mental health. We conducted a 12-month observational study among participants in an HLC that aimed to improve physical activity (PA) and dietary behaviors, and this study examined predictors of completion, and changes in psychological variables, lifestyle behaviors, and physical health indicators. The participants (N = 120, 71% female, mean age = 44 years) reported symptoms of psychological distress (77%) and were obese (77%). No baseline characteristics were found to be consistent predictors of completion (42%). Completers had significant improvements in autonomous motivation for PA (*d* = 0.89), perceived competence for PA (*d* = 1.64) and diet (*d* = 0.66), psychological distress (*d* = 0.71), fruit intake (*d* = 0.64), vegetable intake (*d* = 0.38), BMI among all participants (*d* = 0.21) and obese participants (*d* = 0.34), body fat percentage among all participants (*d* = 0.22) and obese participants (*d* = 0.33), and lower body strength (*d* = 0.91). Fat-free mass and all forms of PA remained unchanged from baseline to 12 months. Hence, there were indications of improvement among completers on psychological variables, lifestyle behaviors, and physical health indicators. The low rate of completion was a concern, and the unchanged levels of PA reflect an important area of focus for future interventions in the context of HLCs.

## 1. Introduction

Physical inactivity and unhealthy diet are behavioral risk factors that contribute to an increased risk of non-communicable diseases (NCDs) and to being overweight or obese—defined by the World Health Organisation (WHO) as an excessive amount of body fat that might impair health, and classified as a Body Mass Index (BMI) ≥ 25 kg/m^2^ [1,2]. Moreover, whereas psychological distress is associated with physical inactivity [3,4] and unhealthy diet [5,6,7], exercise has been found to be effective in treating depression [8] and anxiety [9]. The WHO recommends at least 150–300 min of moderate physical activity (PA), or 75–150 min of vigorous PA throughout the week, or an equivalent combination of moderate to vigorous physical activity (MVPA), along with replacing sedentary time with activity of any intensity, including light PA (LPA). Two days of muscle-strengthening activities are also recommended [10]. Regarding diet, the WHO recommends eating plenty of fruits and vegetables [11], which in Norway is defined as five portions (or 500 g) per day [12]. Adopting a more active lifestyle and healthier diet can promote both physical and mental health [13,14,15], but has proved difficult for many. Therefore, health policies in Norway have shifted towards a focus on disease prevention and health promotion [16,17,18] and established a novel service—Healthy Life Centers (HLCs)—in municipalities throughout Norway. These centers support lifestyle changes and promote physical and mental health through individual guidance and group activities, and are available free of charge for all inhabitants [19]. This study concerns participants in a HLC with a strong emphasis on participant autonomy and possibility for choosing activities that they believed to suit their needs. In this article, the focus of lifestyle change was on PA and dietary behaviors (fruit and vegetable intakes). BMI, body composition (body fat percentage and fat-free mass), and lower body strength were indicators of physical health, and psychological distress was the indicator of mental health.

Research in the context of HLCs is sparse, and it has yielded mixed findings amid low rates of completion (70% at 6 months [20] and 51% at 12 months [21]), which indicates a high rate of drop-out from these services. Participation in HLCs has not been shown to increase PA over prolonged periods of time [20,21] and divergent results have been found regarding changes in BMI [22,23,24], although health-related quality of life (especially mental health) has been shown to improve [21]. The knowledge base generated from research on lifestyle change in settings other than HLCs is also mixed. Numerous studies have shown that increasing PA is possible—yet challenging and depleting—and there is uncertainty about efficacy over time [25,26,27]. Reviews report meaningful changes in diet, operationalized as fruit and vegetable consumption, and suggest that these improvements occur *especially* among at-risk populations [28,29]. A systematic review of meta-analyses reported small effect sizes from interventions that target PA *and* diet, and large effect sizes from interventions that target PA and diet with weight loss as the outcome [30]. Another review found that interventions that address changes in both PA and diet are most effective for weight loss [31]. However, evidence of maintenance of lifestyle change is limited by the low number of studies that have included long-term follow-up and the tendency for treatment gains to diminish over time [32].

Self-determination theory (SDT) is a macro-theory of human motivation that examines the factors that are conducive to the initiation *and* maintenance of health behaviors over time [33]. According to SDT, humans thrive psychologically, physically, and socially when their basic psychological needs for autonomy, competence, and relatedness are supported, and this satisfaction promotes autonomous motivation (an experience of choice and self-endorsement of behavior) and perceived competence (a sense of capability and mastery in action) [34,35,36,37,38,39,40,41], which have been associated with increases in PA and consumption of a healthier diet. It is important to note that health services can support or thwart the satisfaction of basic psychological needs, with consequences for autonomous motivation and perceived competence for a healthy lifestyle [33,42]. Recently, a meta-analysis revealed that SDT-based interventions were able to enhance most health behaviors and physical health outcomes, although changes in psychological health outcomes were less likely to be maintained at follow-up. Moreover, the efficacy of SDT-based interventions was more pronounced for PA than for diet [43]. One study that investigated the role of emotions in motivated behavior found that pleasant emotions can elicit autonomous motivation [44], suggesting that unpleasant emotions might undermine autonomous motivation and lifestyle change.

The services offered by HLCs have the potential to provide support for participants’ basic psychological needs by providing choices about various activities (support for autonomy) from a menu of customized options (support for competence) that are delivered to individuals and/or groups by employees who are experienced in lifestyle counselling (support for relatedness) [19]. Support for basic needs was emphasized in the HLC-offer studied. Theoretically, such support is expected to promote perceived competence and autonomous motivation for lifestyle changes, and the findings from a qualitative study have confirmed this prediction [45]. The satisfaction of basic needs is also likely to promote psychological well-being and reduce psychological distress [43], which are found to be common among those seeking help for lifestyle stress [46]. To date, no research in the context of HLCs has reported on changes in these motivational constructs over 12 months, and no studies have reported on changes in psychological distress, even though emotional struggle has been a prominent theme in qualitative research [47,48]. A few studies have examined changes in PA and BMI, but no studies have reported on changes in diet, body composition, or muscle strength.

Accordingly, the research questions that guided the current study were three-fold. First, what percentage of participants will complete the HLC services at 6 months and 12 months, and will baseline characteristics such as gender, age, ethnicity, education, work status, diagnosis, BMI, psychological distress, autonomous motivation, and perceived competence predict this completion? Second, will autonomous motivation, perceived competence, and psychological distress increase from baseline to 6 months and 12 months? Third, will lifestyle behaviors (PA, fruit and vegetable intakes) and physical health indicators (BMI, body fat percentage, fat-free mass, lower body strength) improve from baseline to 6 months and 12 months?

## 2. Materials and Methods

### 2.1. Study Design

We conducted a 12-month observational study among participants in an HLC intervention at a site that is in one of the largest cities in Norway. The intervention occurred in a real-world setting and was part of the HLC’s standard program; a control group was not established. Measures were obtained at baseline, 6 months, and 12 months. Our goal was to determine if participants will report changes in psychological variables, lifestyle behaviors, and physical health indicators over time, which may lay important groundwork for more rigorous experimental research in the future.

### 2.2. Setting

The HLC services were designed to be individually customized, and participants could choose activities according to what they perceived as necessary to support their process of lifestyle change. All participants received individual guidance, which represented the “lowest level” of intervention, and reflections on activity choice, goal setting, and goal striving were important themes in these consultations. The activities in which participants could engage during the 12-month study period included lifestyle courses (13 weekly sessions), food classes (5 weekly sessions), yoga (for learning relaxation techniques, for body mobility and for body awareness, 8 weekly sessions), and discussion groups for exchanging experiences (8 weekly sessions); walking and training groups were offered continuously. All activities and guidance were provided by educated, experienced, and competent HLC employees. By design, the HLC was intended to be supportive of participants’ basic psychological needs, as each participant was given the opportunity to develop a personally tailored process of lifestyle change (supportive of autonomy and competence) in a caring social context (supportive of relatedness).

### 2.3. Ethics

This study was approved by the Norwegian Data Protection Authority and the Data Protection Officer in the municipality where the HLC is located. All participants provided written, informed consent after receiving information about the purpose and procedures of the study, along with information about how their data would be handled in a secure way.

### 2.4. Participants

All individuals who consulted the HLC during a 20-month period, who met the inclusion criteria, and who agreed to participate were included in the study. Inclusion criteria were being ≥18 years of age (few individuals ≥ 65 years of age consult HLCs, yet none were excluded based on age) with the ability to speak Norwegian and with the intention to become more active and eat healthier (so as to recruit individuals who considered themselves to be inactive). Participants who reported severe and disabling mental illness (and, thus, were not able to attend the HLC without support from external personnel) were excluded. Declining to participate in the study did not affect the services that were offered by the HLC.

Participants who did not attend their scheduled appointments at the HLC were contacted by phone (and then by letter) with an invitation to make a new appointment, after which non-responsive participants were registered as dropouts. A total of 66 (55%) participants provided data at 6 months, and 50 (42%) participants provided data at 12 months.

### 2.5. Measures

All assessments at baseline, 6 months, and 12 months were undertaken at the HLC, and participants completed the self-report measures via an online questionnaire that was aided by SurveyXact (Rambøll Management Consulting, Oslo, Norway). All measures had satisfactory reliability (α ≥ .73). Accelerometers were worn by participants during their daily activities outside the HLC.

**The Health Care Climate Questionnaire** [49] was used to assess perceptions of autonomy support from counsellors at the HLC regarding PA (6 items; e.g., I feel that my counsellor has provided me with choices and options on how I can exercise regularly, or not) and diet (6 items; e.g., I feel that my counsellor has provided me with choices and options on how I can change my diet, or no)]. Responses were made on a 7-point scale from 1 (*not true at all*) to 7 (*completely true*).

**The Treatment Self-Regulation Questionnaire** [50] was used to assess autonomous motivation for both PA and diet, and presented participants with the following stems: “I exercise because…” and “I eat healthy because…”. Participants then rated items on autonomous motivation for PA (6 items; e.g., I want to take responsibility for my own health) and diet (6 items; e.g., I personally believe it is best for my health). Responses were made on a 7-point scale from 1 (*not true at all*) to 7 (*completely true*).

**The Perceived Competence Scale** [50] was used to assess experiences of feeling able to initiate and/or maintain a change around PA (4 items; e.g., I feel confident in my ability to exercise regularly) and diet (4 items; e.g., I am able to meet the challenge of maintaining a healthy diet). Responses were made on a 7-point scale from 1 (*not true at all*) to 7 (*completely true*).

**The Hopkins Symptom Checklist-10** [51] was used to assess experiences of psychological distress during the previous 2 weeks by scoring 10 items relating to symptoms of depression and anxiety. Responses were made on a 4-point scale from 1 (*not at all*) to 2 (*a little*) to 3 (*quite a bit*) to 4 (*extremely*). In the Norwegian version of this measure, a mean value > 1.85 indicates the presence of psychological distress [51,52].

An **Inbody 720** (Body Composition Analyzer, Biospace Co. Ltd., Seoul, Korea) was used to calculate BMI and to measure body composition. Inbody 720 uses direct segmental multifrequency bioelectrical impedance (BIA) for measurements of body components. Assessments were conducted at the HLC by trained personnel, and manufacturer instructions were followed before and during the assessments. Height was measured by a stadiometer and plotted in the Inbody 720 analyzer prior to the assessments. Inbody 720 has been shown to be valid for measuring body composition in general populations and among obese participants [53,54]. BMI, body fat percentage, and fat-free mass (kg) were the variables used in the current study. BMI is categorized as follows: underweight < 18.5 kg/m^2^, healthy weight = 18.5–24.9 kg/m^2^, overweight = 25.0–29.9 kg/m^2^, and obese = ≥30.0 kg/m^2^. The obese category is further classified as follows: obese class 1 = 30.0–34.9 kg/m^2^, obese class 2 = 35.0–39.9 kg/m^2^, and obese class 3 = ≥40.0 kg/m^2^ [2]. Although there are no established cut-off points or normative standards for body fat percentage [55,56], based on limited data, a healthy range for men is considered to be 10–22% and a healthy range for women is considered to be 20–32% [56].

**The ActiGraph GT3X** (ActiGraph GT3X; Pensacola, FL, USA) [57,58] accelerometer was used to measure PA, and it was given to participants during their first appointment at the HLC (also at their 6-month and 12-month assessments) and returned seven days later. Participants were instructed to wear the accelerometer on their right hip during waking hours (except during water activities) for seven consecutive days. Participants recorded a log of their water activities and cycling. The accelerometer was set to record at a sampling rate of 30 Hz in 1 s epochs. Data were included if participants had at least 10 h (600 min) of valid activity recordings per day for at least two days. The mean (SD) number of days with recordings was 5.7 (1.4) at baseline, 6.4 (1.3) at 6 months, and 5.6 (2.1) at 12 months. Non-wear time was defined as more than 60 min of zero counts (spikes for 2 min with counts above 100 were allowed) and was excluded from analyses. Based on Troiano et al. [59], we defined the cut-off points for PA levels using registered counts per minute (cpm) as follows: sedentary = 0–99 cpm, light = 100–2019 cpm, moderate = 2020–5998 cpm, and vigorous = 5999 cpm and higher. Moderate and vigorous forms of PA were combined and reported together as MVPA.

**The 30 s chair-stand test** was used to measure lower body strength. This test has stable reliability and moderately high correlation with maximum weight-adjusted leg-press performance [60]. Although this test was developed for individuals ≥ 60 years old, it was used in the current study because participants were expected to have less strength due to their inactivity. This test was administered by trained personnel at the HLC.

**Fruit and vegetable intakes** were assessed by asking participants how many fruits and vegetables they typically ate every day. Previous research has demonstrated the utility of this approach for monitoring dietary preventive efforts [61].

### 2.6. Sample Size and Statistical Analysis

We conducted power calculations prior to the start of the study, in which a dropout rate of 50% was anticipated based on a combination of clinical experience and previous research [23,62]. A sample size of 120 was found necessary. The full sample size would yield adequate statistical power to reveal small to medium effect sizes (*d* > 0.45 for independent *t*-tests, *d* > 0.26 for dependent *t*-tests, and a difference in percentages of about 25% with groups of equal size). With a sample size of 60, a dependent *t*-test would identify change yielding a *d* = 0.40 (α = .05 and 1 − β = .80) [63,64].

All analyses were conducted using IBM SPSS Statistics 24 (IBM; Armonk, New York, NY, USA). After ensuring that the data did not violate statistical assumptions, we used means (with standard deviations) and frequencies (with percentages) to describe the continuous and categorical variables, respectively. For our first research question, we used independent-samples *t*-tests to examine baseline differences in the continuous variables (age, BMI, psychological distress, autonomous motivation, perceived competence) and cross tabulation with chi-square tests for independence to examine baseline differences in the categorical variables (gender, education, ethnicity, work status, diagnosis) between participants who left the HLC before 6 months or 12 months (dropouts) and those who completed activities in the HLC at 6 months and 12 months (completers). Education was dichotomized as *low education* (primary school and high school) and *high education* (college or university, 3 years and college or university, >3 years). Work status was dichotomized as *able to work* (full time, reduced capacity, partial sick leave, student, partly disabled, and retired) and *unable to work due to health* (sick leave and disabled). For our second and third research questions, we used paired-samples *t*-tests to examine changes in psychological variables (autonomous motivation, perceived competence, psychological distress), lifestyle behaviors (PA, fruit and vegetable intakes), and physical health indicators (BMI, body composition, lower body strength) from baseline to 6 months and 12 months. Changes in BMI and body composition were analyzed among all participants, among participants with BMI ≥ 25.0, and among participants with BMI ≥ 30.0. Unless otherwise stated, the results are presented as mean (SD). Effect sizes (*d*) were defined as follows: small = 0.2, medium = 0.5, large = 0.8, and very large = 1.3 [65]. Effect sizes (*phi*) were defined as follows: small = 0.1, medium = 0.3, and large = 0.5 [66].

## 3. Results

A total of 120 individuals (85 females, 35 males) were included and provided data at baseline. As shown in Table 1, at baseline most participants were female, middle-aged, and of Norwegian ethnicity. A majority of participants were able to work, had lower levels of education, experienced symptoms of psychological distress, were obese, and reported one or more diagnoses.

Of the 120 participants who were recruited at baseline, 66 participants (55%) completed activities in the HLC at 6 months and 50 participants (42%) completed activities in the HLC at 12 months. Figure 1 presents a flow chart of participation in the study from baseline to 12 months. Reasons for dropout included injury, sickness, not finding the activities relevant, lack of time, and pregnancy; 75% of the participants who dropped out did not give a reason for doing so. During the 12-month study period, 39 participants indicated that they took part in lifestyle courses, 22 participants took part in food classes, 20 participants took part in yoga, and 16 participants took part in discussion groups for exchanging experiences. Many participants took part in more than one activity, while a few participants engaged with individual consultations only. All participants were involved in training activities at the HLC and/or in training activities that were self-organized outside the HLC. All participants had individual consultations over the 12-month study period, and the number of consultations ranged from three to eight. Very few participants reported engagement in cycling or water activities, and thus these activities were not included.

### 3.1. Research Question 1: Predictors of Completion at 12 Months

Table 2 presents differences in distribution of baseline categorical and continuous variables between completers and dropouts at 12 months, to illuminate whether the preconditions were different in the two groups.

Chi-square tests for independence revealed no significant differences in gender, education, ethnicity, work status, or diagnosis between completers and dropouts. Independent-samples *t*-tests revealed no significant baseline differences in age, BMI, psychological distress, autonomous motivation, or perceived competence between completers and dropouts. However, the following variables yielded differences that had an effect size ≥ small: ethnicity, work status, diagnosis, and autonomous motivation for diet.

### 3.2. Research Questions 2 and 3: Changes in Psychological Variables, Lifestyle Behaviors, and Physical Health Indicators from Baseline to 6 Months and 12 Months

Completers reported high average levels of autonomy support from counsellors at the HLC regarding PA at 6 months (6.0 (0.9)) and at 12 months (5.9 (1.0)), and regarding diet at 6 months (5.9 (1.0)) and at 12 months (5.8 (1.0)) (maximum scale scores = 7).

Table 3 presents means and standard deviations for the study variables at baseline, 6 months, and 12 months, in addition to measures of the effect size used to quantify the changes in psychological variables, lifestyle behaviors, and indicators of physical health from baseline to 6 months and 12 months.

Regarding psychological variables, there were significant increases in autonomous motivation for PA, perceived competence for PA, and perceived competence for diet. A significant decrease in psychological distress was also found. The effect size for these changes ranged from medium to very large. There was no change in autonomous motivation for diet.

Regarding lifestyle behaviors, there was a significant decrease in sedentary time at 6 months, which was not found at 12 months. Significant increases in fruit and vegetable intakes were also found at 12 months. The effect size for these changes ranged from small to medium. There was no change in LPA or MVPA.

Regarding physical health indicators, there were significant decreases in BMI and body fat percentage among all participants, among participants with BMI ≥ 25.0, and among participants with BMI ≥ 30.0. There was also a significant increase in lower body strength. The effect size for these changes ranged from small to large. Fat-free mass remained unchanged.

## 4. Discussion

This observational study followed participants in an HLC as they strived toward lifestyle change over 12 months. Results revealed that a concerningly high percentage of participants dropped out of the HLC, as only 42% of participants completed HLC services at 12 months. However, completers showed indications of improvements in psychological variables, lifestyle behaviors, and physical health indicators. Levels of PA did not change. We reflect on these findings below.

Across the 12-month study period, 55% of the participants completed activities in the HLC at 6 months, whereas 42% of the participants completed activities in the HLC at 12 months. This may suggest that roughly half of the participants did not succeed at their lifestyle changes, and most participants who dropped out did so without giving notice or reason, and they did not respond to contact attempts from the HLC. There were no predictors of completion, although we encourage caution in the interpretation of these findings due to low statistical power. There were small differences in the variables ethnicity, work status, and diagnosis, and these may have been significant predictors of completion in a larger sample. It is useful to note that low rates of completion have been reported previously with HLCs, although our inability to predict completion stands in some contrast to previous research that has shown that older age and no mental disease are common predictors of completion [20,21,23]. Moreover, Blom et al. [21] found that being Norwegian and having a higher level of education were associated with completion, whereas being on sick leave predicted dropout.

There can be many reasons for dropping out of HLC services, including a lack of time and/or energy to pursue lifestyle changes and frustration from a lack of progress towards one’s goals. Another factor affecting completion may also be the level of positive experiences from attending the HLC. Additionally, the level of psychological distress at baseline was quite high [46], which could have rendered our participants especially vulnerable to dropout considering previous research has shown that completion is predicted by a lack of mental disease [20,21]. The intervention was tailored to support basic needs, which should be beneficial for those who struggle with mental issues. Yet, results suggest that this has not been enough for many of the participants. However, the marked reduction in psychological distress among completers suggest that such an improvement is vital for completion. Indeed, it might be critical to address mental health issues in interventions that support lifestyle changes. Regarding psychological distress, neither autonomous motivation nor perceived competence at baseline predicted completion at 6 months and 12 months, although these variables increased over time among completers. Issues of motivation have been shown to be relevant when predicting dropout from, and commitment to, clinical interventions [67], and motivation is a well-known predictor of maintained lifestyle change [35,36,37,41]. BMI did not predict completion in the current study, but dropout might have resulted from frustration due to a lack of progress in weight reduction and dissatisfaction with one’s body, as completion has been shown to be inversely associated with obesity [68,69].

The current study investigated perceptions of autonomy support and changes in the psychological variables of autonomous motivation, perceived competence, and psychological distress. The HLCs are designed to provide support for participants’ basic psychological needs for autonomy, competence, and relatedness [19]. Completers perceived the intervention to be need supportive, which may explain their reported increases in autonomous motivation for PA, and perceived competence for PA and diet. Autonomous motivation for diet did not increase over time, perhaps due to a ceiling effect at baseline. The decrease in psychological distress was large in magnitude, and the high average levels of psychological distress at baseline raise the question of why some of the participants enrolled in an HLC that was focused on PA and diet rather than on mental health. Speculatively, it is possible that stigma around traditional mental health services acted as a barrier to seeking treatment [70] and that lifestyle interventions are more acceptable and accessible. Without a control group, we cannot conclude that a causal association exists between participation in the HLC and improvements in psychological distress, but it is important to note that previous research has shown that lifestyle interventions tend to yield benefits for mental health [21,71] and to reflect on why psychological distress improved during the 12-month study period. It is possible that participation in the HLC’s activities left participants feeling more optimistic towards and capable of lifestyle changes, and that interacting with others who were facing similar challenges left participants feeling more supported regarding their lifestyle changes. Notably, psychological distress is associated with unhealthy behavior [3,4,5,6,7,8], whereas satisfaction of basic psychological needs is associated with autonomous motivation and better mental health [72]. Additionally, previous research has found that pleasant emotions elicit autonomous motivation [44], perhaps in a reciprocal way. Thus, the improvements in psychological distress, autonomous motivation, and perceived competence that were observed in the current study—due, perhaps, to support for autonomy, competence, and relatedness—may serve to reduce a traditional barrier to the initiation and maintenance of health behavior change [47]. It is important for future research to examine this possibility.

Finally, the current study examined changes in lifestyle behaviors (PA, fruit and vegetable intakes) and physical health indicators (BMI, body composition, lower body strength). Although autonomous motivation and perceived competence for PA increased over time, PA levels did not change during the 12-month study period. This may have been due to the high levels of MVPA at baseline. The WHO recommends at least 150 min of such PA per week [10] (or 21.4 min per day), and thus it is impressive that participants had about 40 min per day at each assessment. As participants may have been overly eager to exert themselves physically at the start of the program, there may have been little room left for improvement. Nevertheless, it is important to note that completers maintained their rather high levels of MVPA over time, as was shown also by Blom et al. [21] (36 min per day of MVPA at baseline, and no change at 15 months) and Samdal et al. [20] (54 min per day of MVPA at baseline, and no change at 6 months) in the context of HLCs. (It is important to note that Samdal et al. [20] used a different type of accelerometer and integrated the data into longer epochs.) The comparatively low levels of LPA and high levels of sedentary time at baseline did not improve by 12 months, which is concerning and suggests that the value of PA throughout the day may not have been communicated well. Even though recommendations are not specific for LPA and sedentary time, apparently the message that “every move counts towards better health” [10] was not adopted by participants. The high levels of MVPA may have undermined the perceived value of increasing LPA. Additionally, the high levels of BMI at baseline may have attenuated the possibility for improvements in PA, as suggested by Ekelund et al. [73]. By contrast, increases in lower body strength were large in magnitude and can be attributed to the high levels of MVPA and/or strengthening exercises not captured by accelerometers, indicating that participants made efforts towards their PA routines. To date, there are no comparable studies on changes in muscle strength, but our findings underscore the importance of adopting a “wide lens” on PA and investigating both changes in PA and consequences of those changes.

There was a small to medium increase in daily intake of fruits and vegetables. Such changes, although significant, do not provide a full account of participants’ diets; rather, they indicate that improvements in diet were made from 2.8 portions to 4 portions [400 g] of fruits and vegetables per day. The WHO considers daily consumption of less than 400 g of fruits and vegetables to be among the top 10 leading causes of death, and an estimated 1.7 million lives can be saved by increasing such consumption to 400 g or more per day [74,75]. Thus, these increases in the current study can be viewed as meaningful and in line with previous findings [28,29]. Nonetheless, the recommendation of five portions per day of fruits and vegetables was not achieved by the participants after one year. Moreover, fruit and vegetable intakes were assessed by two items only. Together, these issues highlight the importance of additional clinical and empirical attention being directed toward this important lifestyle behavior.

There was a small, although significant, decrease in BMI and body fat percentage, which, coupled with the increase in fruit and vegetable intake, may indicate an overall improvement in the consumption of a healthy diet. Fat-free mass did not change, which suggests that the decreases in BMI were a function of reductions in fat mass. The effect sizes for changes in BMI and body fat percentage were larger among overweight and obese participants. Although participants were not recruited into a weight-reduction intervention, a majority of participants were obese and likely had intentions to reduce and/or stabilize their weight. From this perspective, a small effect size for reductions in BMI and body fat percentage over one year is sobering, although there may be some clinical relevance to these findings given that average BMI among obese participants decreased from 36.1 (obese class 2) to 34.5 (obese class 1). Indeed, relative to obese class 2, both men and women in obese class 1 have an increased life expectancy of more than two years [2]. Our findings are aligned with Sweet and Fortier [30] systematic review that reported small effect sizes from interventions that aim to improve PA *and* diet (such as HLCs), and our findings are contrary to the large effect sizes that were found among interventions that target weight loss. The small decreases in BMI and body fat percentage might be due to complexities around eating regulation and pathological eating [35]. Qualitative findings from the same sample of participants illustrated the importance, and difficulty, of coping with stress and unpleasant emotions using healthy strategies, rather than using food to regulate mood [45]. Though speculative, it is possible that the decrease in psychological distress along with the increases in autonomous motivation and perceived competence for diet might indicate the beginning of more healthy dietary regulation.

The intent of an HLC is to empower individuals to move towards sustainably healthier lives, and it seems that the HLC that was examined in the current study was partly successful in doing so. The health services were perceived as need supportive, and completers reported improvements in the psychological variables of autonomous motivation, perceived competence, and psychological distress. The findings with regard to lifestyle behaviors were less clear as fruit and vegetable intakes increased over time but PA did not, which is inconsistent with previous research from SDT [34,38] but consistent with previous research from HLCs [20,21]. As levels of MVPA were rather high at baseline, it may have been unrealistic to expect further increases. On the other hand, the lack of increase in LPA was discouraging. Regarding physical health indicators, the effect sizes for changes in lower body strength were large, whereas the effect sizes for changes in BMI and body fat percentage were small. Of course, it is important to bear in mind that more than half of the participants left the HLC during the 12-month study period, and it is critical for future research—and the HLC—to focus on ways to improve rates of completion.

### Limitations and Directions for Future Research

This study has several limitations. First, without an experimental design and a control group, we cannot conclude that a causal association exists between participation in the HLC and the changes that were observed. It is important for future research not only to test for causality, but also to examine directionality between changes in psychological variables, lifestyle behaviors, and physical health indicators over time in the context of an HLC. Nor was it possible to “tease apart” various components of the intervention. Indeed, we simply observed changes over time, which may be attributed to factors that are also unrelated to the HLC. Second, participants were recruited from one HLC only, which limits the generalizability of our findings. Nonetheless, sample characteristics were similar to those of other recent HLC studies [20,21], and HLCs in Norway employ a similar structure. Third, it is possible that our sample was too small and/or homogenous, which may have increased the risk of type 2 error and resulted in a lack of statistical power to detect demographic factors that may have been significant predictors of completion. Unfortunately, the resources necessary to recruit a larger, more diverse sample were simply not available. Finally, only 42% of the participants completed activities in the HLC at 12 months, and such attrition may have biased the results in a positive direction given that completers tended to improve over time. Nevertheless, completers and dropouts were roughly equivalent at baseline, which mitigates some concern here. It is important for future research to develop more understanding of reasons for dropout and examine how attrition might affect the results. Moreover, it is important for future research to maximize completion in the context of an HLC.

## 5. Conclusions

The findings suggest that the activities provided by HLCs are potentially beneficial in some ways for those who complete services at 12 months since autonomous motivation and perceived competence increased, and psychological distress decreased. These improvements in psychological constructs can indicate an improved foundation for continuing the strive towards a healthy lifestyle. Findings for lifestyle behaviors were less consistent. Completers reported meaningful improvements in diet, but their levels of PA remained unchanged. Although rather high levels of MVPA were maintained, greater effort may have been made to reduce sedentary time and increase levels of LPA. Moreover, physical health indicators, namely, BMI, body composition, and lower body strength, tended to improve. An issue of concern was the high rate of dropout, which could have biased the results in a positive direction. More knowledge is needed about how the service could be adapted or strengthened to encourage a higher level of retention by investigating reasons for dropout Indeed, attrition might be attenuated by a focus on mental health issues in the context of the intervention. As support for the basic psychological needs is likely to be important in this process, it is critical to assess perceptions of need support more systematically and earlier in the intervention.

## Figures and Tables

**Figure 1 ijerph-19-05167-f001:**
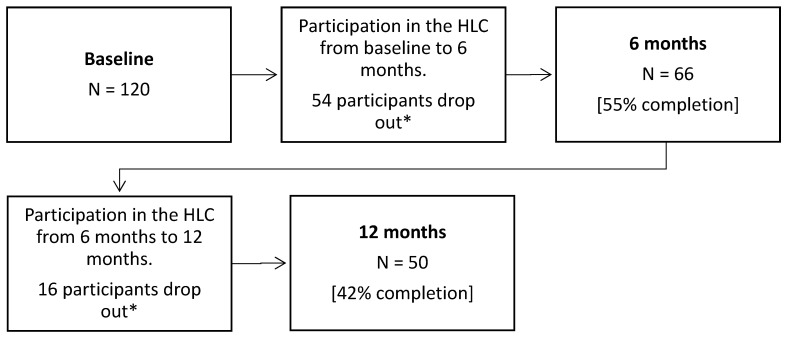
Flow chart of participation in the study from baseline to 12 months. * A majority of the participants who dropped out did so without giving notice or reason, and they did not respond to contact attempts from the HLC. Reasons for dropout included injury, sickness, not finding the activities relevant, lack of time, and pregnancy.

**Table 1 ijerph-19-05167-t001:** Sample characteristics at baseline (*N* = 120).

**Gender: *n* (%)**	
**Female**	85 (71%)
**Male**	35 (29%)
**Age: Mean (SD)**	44 (14)
**Ethnicity: *n* (%)**	
**Norwegian**	102 (85%)
**Other**	18 (15%)
**Education: *n* (%)**	
**Primary school**	20 (17%)
**High school**	54 (45%)
**College or university (3 years)**	20 (17%)
**College or university (>3 years)**	26 (21%)
**Work status: *n* (%)**	
**Full time**	24 (20%)
**Reduced capacity**	20 (17%)
**Partial sick leave**	6 (5%)
**Sick leave**	30 (25%)
**Student**	13 (11%)
**Partly disabled**	1 (1%)
**Disabled**	22 (18%)
**Retired**	4 (3%)
**Diagnosis (self-reported): *n* (%)**	
**Yes**	73 (61%)
**No**	47 (39%)
**Body mass index (kg/m^2^): Mean (SD)**	34.5 (6.2)
**Body mass index classification: *n* (%)**	
**BMI < 25**	8 (7%)
**BMI 25–29**	19 (16%)
**BMI 30–34**	42 (35%)
**BMI ≥ 35**	50 (42%)
**Psychological distress: Mean (SD)**	2.4 (0.6)

**Table 2 ijerph-19-05167-t002:** Differences in distribution of baseline categorical and baseline continuous variables between completers and dropouts at 12 months.

Categorical Variables	Dropouts ^1^	Completers ^2^	*p* Value ^3^	Effect Size (*phi*)
Gender: % at baseline				
Female	58.8	41.2	1.0	.015
Male	57.1	42.9		
Ethnicity: % at baseline				
Norwegian	55.9	44.1	.30	.118
Other	72.2	27.8		
Education: % at baseline				
Low	60.8	39.2	.61	.064
High	54.3	45.7		
Work status: % at baseline				
Able	51.5	48.5	.12	.159
Unable	67.3	32.7		
Diagnosis: % at baseline				
Yes	53.4	46.6	.24	.124
No	66.0	34.0		
Continuous variables				Effect size (*d*)
Age (SD)	44.0 (13.5)	43.4 (14.4)	.82	.04
Autonomous motivation (SD)				
Diet	6.1 (1.0)	5.9 (0.9)	.20	.21
Exercise	4.8 (1.3)	4.7 (1.2)	.50	.08
Perceived competence (SD)				
Diet	4.0 (1.2)	4.0 (1.2)	.84	.00
Exercise	3.0 (1.5)	3.0 (1.7)	.89	.00
Psychological distress (SD)	2.4 (0.6)	2.4 (0.6)	.85	.00
Body Mass Index ^4^ (SD)	34 (6.4)	35 (6.1)	.46	.16

^1^ n = 50. ^2^ n = 70. ^3^ The *p* values from Yates’ Correction for Continuity Test were derived from a 2 × 2 table. ^4^ n = 50 for completers and n = 69 for dropouts.

**Table 3 ijerph-19-05167-t003:** Means and standard deviations for the study variables at baseline, 6 months, and 12 months.

Variable	Completers at 6 Months		Effect Size	Completers at 12 Months		Effect Size
	Baseline ^1^ Mean (SD)	6 Months ^1^ Mean (SD)	*d*	Baseline ^2^ Mean (SD)	12 Months ^2^ Mean (SD)	*d*
Autonomous motivation						
Physical activity	4.7 (1.3)	6.2 (1.0) ***	1.30	4.8 (1.3)	5.8 (0.9) ***	0.89
Diet	6.0 (0.9)	6.0 (0.9)	0.00	6.1 (1.0)	6.0 (0.9)	0.10
Perceived competence						
Physical activity	2.8 (1.4)	4.8 (1.5) ***	1.40	3.0 (1.5)	5.3 (1.3) ***	1.64
Diet	3.9 (1.1)	4.6 (1.5) ***	0.53	4.0 (1.2)	4.9 (1.5) ***	0.66
Psychological distress	2.4 (0.7)	1.9 (0.7) **	0.71	2.4 (0.6)	1.9 (0.8) **	0.71
Physical activity						
LPA min/day	83.9 (25.5)	80.5 (35.1)	0.11	80.5 (27.4)	76.0 (31.0)	0.15
MVPA min/day	45.5 (21.1)	44.4 (17.7)	0.06	43.5 (20.3)	41.2 (20.0)	0.11
SED min/day ^3^	743.2 (70.9)	719.5 (98.1) *	0.28	749.1 (74.5)	728.4 (84.8)	0.26
Fruit intake	---	--- ^4^	---	1.3 (1.2)	2.1 (1.3) ***	0.64
Vegetable intake	---	--- ^4^	---	1.5 (1.2)	1.9 (0.9) *	0.38
30-s chair-stand test	14.4 (4.0)	16.9 (4.7) ***	0.57	14.1 (3.9)	18.6 (5.8) ***	0.91
Body mass index ^5^ (kg/m^2^)						
BMI All	33.9 (6.0)	33.3 (5.8) ***	0.10	33.5 (6.1)	32.3 (5.3) **	0.21
BMI ≥ 25	35.0 (5.0)	34.3 (4.9) ***	0.14	34.6 (5.0)	33.3 (4.3) **	0.28
BMI ≥ 30	36.5 (4.5)	35.8 (4.5) **	0.15	36.1 (4.7)	34.5 (4.7) **	0.34
Body composition ^5^						
Body fat %, All	41.5 (9.2)	40.5 (9.1) *	0.11	41.1 (10.0)	38.9 (9.6) *	0.22
Body fat %, BMI ≥ 25	43.4 (6.2)	42.3 (6.6) ***	0.17	43.4 (6.6) **	41.0 (6.8)	0.36
Body fat %, BMI ≥ 30	43.5 (6.9)	42.7 (6.6) ***	0.12	43.0 (7.6)	40.6 (7.1) **	0.33
Fat-free mass, All	55.8 (12.7)	55.7 (12.2)	0.01	55.4 (12.1)	55.4 (10.8)	0.00
Fat-free mass, BMI ≥ 25	56.0 (12.9)	55.9 (12.4)	0.01	55.4 (12.3)	55.3 (10.9)	0.01
Fat-free mass, BMI ≥ 30	60.2 (16.6)	59.4 (14.1)	0.05	59.9 (17.4)	59.1 (14.2)	0.05

^1^ n = 66, except for PA (n = 59). ^2^ n = 50, except for PA (n = 44) and 30 s chair-stand test (n = 49). ^3^ n = 59 at 6 months and n = 43 at 12 months due to removal of one outlier. ^4^ There was an error in the collection of fruit and vegetable intakes at 6 months. ^5^ At 6 months, n for All = 64, n for BMI ≥ 25.0 = 59, n for BMI ≥ 30.0 = 52. At 12 months, n for All = 48, n for BMI ≥ 25.0 = 44, n for BMI ≥ 30.0 = 40. LPA = Light PA. MVPA = Moderate to vigorous PA. SED = Sedentary time. Cohen’s *d* was the measure of effect size. * *p* < .05, ** *p* < .01, *** *p* < .001.

## Data Availability

The data used in the current study are not publicly available, yet they can be made available [anonymized] by CHS upon reasonable request so long as permission for data storage is applicable.

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
