# Peer review of "Motivation and Lifestyle-Related Changes among Participants in a Healthy Life Centre: A 12-Month Observational Study"

_ijerph, 2022, doi:10.3390/ijerph19095167_

Round 1

Reviewer 1 Report

A Healthy Life Centre (HLC) is an interdisciplinary primary health care service which offers effective, knowledge-based programs and methods for people with, or in high risk of disease, who need support in health behavior change and in coping with health problems and chronic diseases.

Moreover, overweight and obesity is a big problem for the health of human beings in Norway, as it is in the rest of Europe.

The interesting paper finds that the activities provided by HLCs are potentially beneficial in some ways for those who complete services at 12 months since autonomous motivation and perceived competence increased, and psychological distress decreased.

The HLC is part of the public health care service in the municipality. HLC programs have a patient oriented approach and aim at strengthening the individual's control of his or her own health (empowerment). As a minimum HLCs offer various exercise groups, and individually or group based counselling or courses.

The authors are advised to broaden the focus and interest of the paper to include the problem of occupational health and safety, as well as other important causes of disease. It should include some citations such as doi: 10.3989/ic.09.070; doi: 10.3989/ic.11.040; doi:10.3390/ijerph17031107; doi.org/10.3989/ic.12.035 among others.

It is suggested that the authors include a graph that clearly shows the results obtained in order to improve the work.

The explanatory text of Figure 1 should be placed below the figure.

Author Response

Thank you for talking your time to read the manuscript, and suggestion valuable improvements.

Comment 1

The authors are advised to broaden the focus and interest of the paper to include the problem of occupational health and safety, as well as other important causes of disease. It should include some citations such as

doi: 10.3989/ic.09.070; doi: 10.3989/ic.11.040; doi:10.3390/ijerph17031107; doi.org/10.3989/ic.12.035 among others.

Response:

Thank you for suggestions, but the suggested articles found on these doi’s are on work safety and as the scope for our article is on lifestyle-change we do not find them eligible for the context.

Comment 2:

It is suggested that the authors include a graph that clearly shows the results obtained in order to improve the work.

Response:

We have included tables to show the results, these are necessary to show all the values on the variables. We find it challenging to find “space” for additional graphs for each result as the scoring range varies, implying we would need 8 graphs, with several pillars on each. This could be made for supplementary files if necessary.

Comment 3:

The explanatory text of Figure 1 should be placed below the figure.

Response:

We have moved the text of Figure 1, see line 276, page 6.

Reviewer 2 Report

This is an important study reporting on the results of a behavioral intervention in Healthy Life Centres in Norway.
However, some modifications are required.

Regarding exercise, there are 8 weekly sessions of yoga, which is a lot. This is not even mentioned in the WHO physical activity guidelines. Please explain the purpose of this menu.

The period of enrollment of the subjects is not mentioned, is it affected by COVID-19?

The row for Physical activity in table3 is out of alignment.

Author Response

Thank you for talking your time to read the manuscript, and suggestion valuable improvements.

Comment 1:

Regarding exercise, there are 8 weekly sessions of yoga, which is a lot. This is not even mentioned in the WHO physical activity guidelines. Please explain the purpose of this menu.

Response:

Yoga is a part of the offers at HLC with the intention to provide the participants techniques for relaxation, for body mobility and for body awareness. This is now included in the article, see line 130, page 2.

Comment 2:

The period of enrolment of the subjects is not mentioned, is it affected by COVID-19?

Response:

Thank you for a good point, however the enrolment was prior to COVID-19. We have commented this in the article, see line 156, page 3.

Comment 3:

The row for Physical activity in table3 is out of alignment.

Response:

We have tried to align table 3, but we struggle to correct it and think it might be du to the setup used by the journal? In the original word manuscript, we find the table aligned. The variables we find out of line in the IJERPH setup are:

  • Autonomous motivation, both for diet and PA.
  • BMI
  • Body fat %, BMI ≥ 30
  • Fat free mass

This manuscript is a resubmission of an earlier submission. The following is a list of the peer review reports and author responses from that submission.

Round 1

Reviewer 1 Report

Comments to the Author

                This study evaluated the effects of HLC on physical activity and psychological factors, and found that psychological variables, lifestyle behaviours, and physical health indicators improved.

The characteristics at baseline shown in table1 should be written in "result", otherwise it is difficult for readers to understand.

Major comments:

  1. The authors argue the usefulness of this study with respect to the activity provided by HLC. However, there was no control group in this study, making it difficult to emphasize the usefulness of HLC. The study design should be changed or listed as a limitation.
  2. It is unclear what happened to the interrupters in this study afterwards. Considering the high discontinuation rate, it is necessary to examine the possibility that the subjects who were able to continue in this study are a good group to change their treatment behavior.

Minor comments:

  1. The characteristics at baseline shown in table1 should be written in "result", otherwise it is difficult for readers to understand.
  2. Is the "lower body strength" in Table 3 the 30-s chair-stand test? I think it would be better to use The 30-s chair-stand test and include the units.

Reviewer 2 Report

An original article was presented to me for review, the purpose of which was to discuss the course and conclusions of the observational study. Despite the efforts of the authors, in my opinion the work does not bring anything new to the world of science and does not translate into everyday life. It does not bring anything new to the subject of metabolic disorders, does not indicate eating habits, the type and frequency of physical activity undertaken, and the mechanisms influencing the "fight" with obesity (a significant number of respondents had a BMI> 30 kg / m2).

The introduction of the work is unnecessarily long and the authors should shorten it to the most important information. The aim of the work does not present anything innovative, it lacks translation into clinical practice. The methodology of work is unnecessarily long and requires shortening to the most important issues. As the authors rightly mention in the "work limitations" section, it lacks a control group that would present the possibility of referencing and comparing the results. Moreover, the age range of the respondents is very wide - which is reflected in the results of the study. The diet was not assessed, the energy and nutritional supply was not assessed, and there was residual information in the study, eg on increasing the consumption of vegetables and fruit .. which is insufficient in the assessment of nutritional modifications of the people participating in the study. The "Results" section is illegible, please rewrite it.

In addition, there are numerous punctuation errors, double spaces, quotations should be placed in square brackets, there are different fonts in the paper, which suggests copying and pasting text, some parts of the text are also unnecessarily bold. The conclusions of the thesis are a summary, not a conclusion, and should be redrafted. In my opinion, the work does not constitute any innovation and does not differ from the generally possessed knowledge, so I apply for its rejection in the review process.